# Interleukin (IL)-9 Supports the Tumor-Promoting Environment of Chronic Lymphocytic Leukemia

**DOI:** 10.3390/cancers13246301

**Published:** 2021-12-15

**Authors:** Laura Patrussi, Nagaja Capitani, Cosima T. Baldari

**Affiliations:** Department of Life Sciences, University of Siena, Via Aldo Moro 2, 53100 Siena, Italy; baldari@unisi.it

**Keywords:** IL-9, CLL, tumor microenvironment, cytokine, hematologic malignancies

## Abstract

**Simple Summary:**

Interleukin 9 (IL-9), a soluble factor secreted by immune cells, has been found in several tumor niches where, depending on the specific tumor type, it either promotes or counteracts tumor development. Recently, IL-9 has been implicated in the development of chronic lymphocytic leukemia, although the underlying molecular mechanism remains unknown. Here, we summarize the current knowledge concerning the roles of IL-9 in disease, with a focus on its implication in the pathogenesis of chronic lymphocytic leukemia.

**Abstract:**

Interleukin (IL)-9 is a soluble factor secreted by immune cells into the microenvironment. Originally identified as a mediator of allergic responses, IL-9 has been detected in recent years in several tumor niches. In solid tumors, it mainly promotes anti-tumor immune responses, while in hematologic malignancies, it sustains the growth and survival of neoplastic cells. IL-9 has been recently implicated in the pathogenesis of chronic lymphocytic leukemia; however, the molecular mechanisms underlying its contribution to this complex neoplasia are still unclear. Here, we summarize the current knowledge of IL-9 in the tumor microenvironment, with a focus on its role in the pathogenesis of chronic lymphocytic leukemia.

## 1. Introduction

Within the tumor microenvironment cytokines, soluble proteins that mediate cell-to-cell communication are considered as dual players. They target immune and non-immune cells expressing their receptors and activate signaling cascades that stimulate anti-cancer responses. Since the discovery of the activities of the pro-inflammatory cytokines Interferon (IFN)-α and Interleukin (IL)-2 against a number of malignancies, we have witnessed an exponential increase in the number of clinical trials addressing both the safety and the efficacy of cytokine-based drugs [1]. By contrast, cytokines can, in some instances, act as potent tumor-promoting agents. Chronic signaling elicited by a number of cytokines was found to be associated to tumorigenesis in a variety of mouse models as well as in human diseases [2]. Cytokines act on tumor cells through downstream signaling mediators to support cancer cell proliferation, survival, and metastatic dissemination [3]. Moreover, they can act extrinsically on other cells within the complex tumor microenvironment, supporting angiogenesis and the tumor evasion of immune surveillance [4]. Importantly, cytokines also modulate cancer cell sensitivity to anti-cancer drugs, thereby contributing to protection from cell death [5].

The relationship between cancer and the tumor microenvironment has long been a confounding issue. Although cells surrounding or recruited to the tumoral milieu have the weapons required to target cancer cells, they are often rewired to tumor-promoting cells [6,7]. Subsets of immune and non-immune cells have been identified in the last two decades as drivers of neoplastic progression: tumor-associated macrophages, neutrophils, myeloid-derived suppressor cells, regulatory T cells (Treg) [8], and cancer-associated fibroblasts [9]. Many of these cells act through the secretion of cytokines, including tumor necrosis factor (TNF)-α [10], transforming growth factor (TGF)-β [11], IL-1 [12], IL-6 [5,13], IL-9 [14,15], and IL-10 [16]. Additionally, in solid tumors that are breast, colon, lung, and kidney cancer, and in hematologic malignancies, tumor cells themselves secrete cytokines to sustain a pro-tumorigenic inflammatory loop [17,18]. Regardless of their cellular source, these immune mediators control the efficient communication between neoplastic cells and bystander cells, and guide the tumor microenvironment to establish a favorable milieu to support tumorigenesis.

A profound modulation of the tumor microenvironment characterizes chronic lymphocytic leukemia (CLL), with altered functions of innate and adaptive immune elements and non-immune cells that favor leukemia onset and evolution and affect therapeutic responses [19]. Among these elements, secreted cytokines stand out through their ability to hamper tumor-directed immune responses and to consequently induce an immunosuppressive pro-survival environment for tumor cells. CLL B cells secrete a large variety of cytokines (extensively reviewed in [17]), which contribute to the altered cytokine balance observed in this leukemia and can be related to the clinical course of the disease [17], both by supporting the growth of the leukemic clone [20] and by hampering apoptotic programs [21]. IL-9, one of the most recent entries in the list of cytokines with effects on CLL, is secreted by several immune cells in many disease contexts where it shows both anti-tumoral [22] and pro-tumoral effects [23] that depend on the specific tumor subtype. The molecular mechanisms underlying the pro-tumoral activities of IL-9 in CLL pathogenesis are still a matter of debate and deserve to be discussed. In this review, we will examine, in detail, IL-9 and its functions as a soluble mediator of immune responses, focusing on its role in the pathogenesis of CLL.

## 2. The Intracellular Signaling Pathways Activated by IL-9

Identified in the late 1980s, the pleiotropic cytokine IL-9 was initially found to act as a growth factor for T lymphocytes and mast cells [24]. In the next decade, it became clear that IL-9 is secreted by a number of cell subsets, including the effector T helper (Th) 2 and Th17 cells, regulatory T cells (Tregs), the type 2 innate lymphoid cells (ILC2s), and natural killer (NK) T cells [24]. A milestone in our understanding of IL-9 biology was reached in 2008 when Veldohen and colleagues reported that a new CD4^+^ T-cell subset, named Th9, that differentiates starting from either naïve T cells in the presence of IL-1β, IL-4, and TGF-β, or from Th2 cells in the presence of TGF-β alone [25], was endowed with the specific ability to secrete high amounts of IL-9 [26]. Recently, other cell populations were found to secrete IL-9, including Tc9 [27], a cytotoxic T cell (CTL) population that differentiates in a microenvironment enriched in IL-9 and that shows a modified expression profile of typical CTL molecules, such as the lytic enzyme granzyme B, the transcription factors Eomes and T-bet, and the pro-inflammatory cytokine IFN-γ [28]. In human peripheral blood, one of the major sources of IL-9 is represented by Vδ2 T cells, the main cell subpopulation of the γδ T cell subset [29]. Indeed, in the presence of TGF-β and IL-15, antigen-stimulated Vδ2 T cells secrete large amounts of IL-9 [30]. Finally, IL-9-producing mucosal mast cells (MMC9), distributed around the microvasculature of the intestinal mucosa [31], participate in allergic diseases and promote food allergy mediated by IgE [32,33], mainly by secreting a variety of cytokines, including IL-9 [34].

As a member of the γ-chain family of cytokines, IL-9 binds to IL-9R, a heterodimeric surface receptor composed of a common γc chain and an IL-9Rα-specific chain that provides the ligand-binding domain [35]. The two receptor monomers associate preferentially following IL-9 binding, as demonstrated by the fact that in the absence of IL-9, only small amounts of IL-9Rα (approximately 25%) are associated with the γc subunit, while in the presence of IL-9, the percentage of heterotypic receptor complexes increases [36], indicating a marked dimerization activity operated by the cytokine ligand. The two molecular components of the IL-9 receptor play distinct roles in intracellular signaling. Upon IL-9 binding, IL-9Rα and the γc chain form a heterocomplex that undergoes conformational changes that allow recruitment of the intracellular tyrosine kinase Janus kinase (JAK) 1 to the consensus intracellular membrane-proximal BOX1 motif of IL-9Rα [37], while the γc chain interacts with JAK3 (Figure 1) [38]. IL-9-induced receptor dimerization then promotes the cross-phosphorylation of JAK1 and JAK3 [39]. Although this upstream signaling module is shared by all other members of the γc chain family of cytokines (interleukins 2, 4, 7, 15 and 21), the downstream signaling is cytokine-specific. The IL-9R-associated phosphorylated forms of JAK1 and JAK3 mediate the phosphorylation of receptor tyrosine residues, which, in turn, act as docking sites for downstream signaling molecules containing Src homology 2 (SH2) domains. These include the transcription factors Signal Transducers and Activators of Transcription (STAT), the Insulin Receptor Substrate (IRS), and adaptors mediating the activation of the Mitogen-Activated Protein Kinase (MAPK) pathways [40]. A single phosphorylated tyrosine residue (tyrosine 367) in the cytoplasmic tail of IL-9Rα transduces IL-9 signaling to STAT proteins [39]. The amino acids flanking tyrosine 367 specificate binding to STAT1, STAT3, and STAT5, leading to the formation of STAT1 homodimers, STAT5 homodimers, and STAT1–STAT3 heterodimers [41], which then translocate to the nucleus to initiate de novo gene expression (Figure 1). In hematopoietic cells, IL-9 was also found to activate IRS-1 and IRS-2 [42], large molecules containing both individual residues and domains that mediate their interaction with signal transduction components, including a protein tyrosine-binding (PTB) domain and several phosphorylatable serine, threonine, and tyrosine residues [43]. Following IL-9 stimulation, JAK1 associates with and phosphorylates IRS-1 [27], which, in turn, interacts with SH2-containing signaling proteins such as the p85 regulatory subunit of Phosphatidylinositol-3 Kinase (PI3-K). Active PI3-K then activates downstream signaling molecules such as Akt, which, in turn, phosphorylates Bcl2-associated Agonist of cell Death (BAD), thereby preventing apoptosis [40] (Figure 1). Weak activation of the MAPK pathway was also reported in lymphoid and mast cell lines stimulated with IL-9 [44]. The adaptors SH2-domain-containing (Shc) and growth factor receptor-bound-protein 2 (Grb2) both participate in this signaling pathway, which leads to the activation of Son of Sevenless (SOS), the GTP exchange factor for the small GTPase Ras, and to the activation of the MAPKs ERK1/2. How Shc and/or Grb2 are recruited to the IL-9 receptor has, as yet, not been elucidated, although a role for additional adaptors was proposed [40]. This signaling module was recently found to be related to the pathogenesis of pediatric T-cell acute lymphoblastic leukemia (T-ALL). Ksionda and colleagues reported that overexpression of the Ras Guanyl exchange factor (GEF) RasGRP1 (Ras guanine nucleotide-releasing protein 1) in T-ALL cells makes them highly sensitive to IL-9, which is, therefore, able to strongly stimulate the Ras-mediated signaling pathways, enhancing leukemic cell proliferation and survival [45] (Figure 1).

As for all signaling modules, the IL-9-dependent signaling pathways need to be negatively regulated through inhibitory mechanisms. Suppressor of Cytokine Signaling (SOCS) 3 [46], Protein Inhibitors of Activated STATs (PIAS), and the SH2-containing phosphatase SH-PTP2 [47] hamper IL-9 signaling by (i) blocking the activation of the STATs (SOCS3), (ii) impairing binding of STAT dimers to their specific DNA target sequences (PIAS), and (iii) dephosphorylating the receptor phosphatase SH-PTP2 [48]. Downregulation of the IL-9-bound surface IL-9R followed by polyubiquitination and proteasomal degradation of both IL-9Rα [49] and the γc chain [50] was also previously reported as a means of definitely turning off IL-9 signaling [40].

## 3. Implications of IL-9 Secretion in Disease

IL-9Rα is expressed on hematopoietic cells, which include Th17, Treg, CTLs, B cells, mast cells, and dendritic cells (DCs) [51]. It is also expressed on non-hematopoietic cells such as airway and intestinal epithelial cells, smooth muscle cells, and keratinocytes [52]. Due to this broad expression pattern, a variety of cell types are sensitive to IL-9 secretion in the microenvironment with effects (extensively reviewed in [51]) that include the following: (i) the stimulation of Th17 differentiation and proliferation; (ii) the enhancement of the suppressive activities of Treg; (iii) the modulation of CTL cytotoxicity; (iv) the activation and proliferation of mast cells, ILCs, and DCs; and (v) the promotion of memory B cell development and antibody-dependent responses [51]. Moreover, IL-9-stimulated cells, in turn, secrete cytokines that exert both autocrine stimulation and feedback stimulatory loops on IL-9-producing cells themselves [51].

Taking into consideration the plethora of IL-9 targets and their multiple and multifaceted roles in immune responses, the effects of IL-9 release are so different that they can even be opposed. Depending on the disease context, IL-9 exerts either stimulatory or suppressive effects on immune responses. IL-9 was initially studied in allergic diseases, where it exerts a detrimental pro-inflammatory activity by promoting the expression of the Th2-related chemokines CCL17 and CCL22, known to be associated with allergic inflammation [53]. Moreover, mast cell-derived IL-9 enhances the susceptibility to IgE-mediated experimental food allergy [32,33]. Th9 cells, through their IL-9-elevating activity, also contribute to the pathogenesis of autoimmune-related diseases, such as rheumatoid and psoriatic arthritis, systemic vasculitis, systemic lupus erythematosus, and systemic sclerosis [52,54]. The overexpression of IL-9 and IL-9Rα in interstitial fluids and tissues isolated from patients with autoimmune diseases has been related to the degree of tissue inflammation [52,55,56]. It is noteworthy that IL-9 expression also ameliorates the outcomes of some types of autoimmune diseases. Notably, it exerts an anti-inflammatory activity in both multiple sclerosis and experimental autoimmune encephalomyelitis (EAE), inflammatory diseases of the central nervous system whose pathogeneses are mainly related to the activity of Th17 cells [57]. In these diseases, IL-9 suppresses the secretion of Granulocyte-Macrophage Colony-Stimulating Factor (GM-CSF) by CD4^+^ T cells, thereby reducing autoimmune neuroinflammation [58]. IL-9 has also a remarkable implication in pathogen clearance. It was reported that it helps in the clearing of parasites by promoting the recruitment and proliferation of mast cells [24]. Moreover, by markedly decreasing the production of the inflammatory mediators TNF-α, IL-12, and IFN-γ and concomitantly inducing the production of the anti-inflammatory cytokine IL-10, it confers resistance to infections of the lethal pathogen *Pseudomonas aeruginosa* [59].

## 4. The Pro- and Anti-Tumoral Functions of IL-9

Wan and colleagues recently defined IL-9 as a “double-edged sword” in tumor immunity [25]. On the one hand, it promotes tumor development by enhancing tumor cell proliferation and blocking their apoptotic program; on the other hand, it mediates anti-tumoral immunity by fueling both adaptive and innate immune responses [25]. The current idea is that IL-9 exerts opposite effects on tumor development according to the type of neoplastic cell and to its microenvironmental niche [17]. In the majority of solid tumors, among which melanoma stands out [60], IL-9 acts as an anti-tumoral factor both by promoting apoptosis of tumor cells and by activating innate and adaptive anti-tumoral immunity. Th9 cells fight against tumors thanks to their ability to secrete high amounts of IL-9, as demonstrated by the fact that IL-9 deletion abolishes the Th9-mediated anti-tumor effects [25,28,61]. In a mouse model of pulmonary melanoma, Th9 cells promote the secretion of CCL20 by epithelial cells, thereby promoting the CCR6-dependent recruitment of DCs to the tumor microenvironment, with subsequent tumor antigen delivery to the draining lymph nodes and CD8^+^ T cell priming, eventually triggering a potent CTL-killing activity [61]. IL-9R^−/−^ mice show increased tumor growth, while, on the other hand, injection of recombinant mouse IL-9 into melanoma-bearing mice inhibits tumor growth [62]. The fact that melanoma growth is hampered by tumor-specific Th9 cell administration, an effect reverted by anti-IL-9 antibodies [62], supports a major anti-tumoral role of IL-9 in this type of cancer. In addition, Th9 cells can also directly kill tumor cells by secreting granzyme B. Pharmacological inhibition of granzyme B activity significantly attenuates the cytotoxic activity of Th9 cells against B16F10 melanoma cells [62]. Th9 cell-derived IL-9 and IL-21 further enhance the ability of CTL and NKT cells to secrete IFN-γ, thereby promoting tumor cell killing [63]. An exception is represented by metastatic lung cancer, a solid tumor where Th9 and Th17 lymphocytes induce metastatic spreading through IL-9 secretion, thereby strongly favoring tumor metastasis [64].

IL-9 also exerts anti-tumoral activity in gastric cancer, as demonstrated by the fact that it inhibits both the proliferation and migration of the gastric cancer cell line SGC-7901 in vitro [65]. Moreover, in a cohort of 453 gastric cancer patients, high IL-9 expression was found to be associated with increased numbers and elevated killing activities of CD8^+^-infiltrating T lymphocytes (TILs), enhanced efficacy of anti-programmed cell death 1 (PD-1) immunotherapy based on the monoclonal antibody Pembrolizumab, and increased overall survival [22]. IL-9 also exerts a strong anti-tumor response in colon cancer [66]. Notably, IL-9 quantification by immunohistochemistry and quantitative real-time PCR in tissue specimens of colon cancer patients showed a strong correlation between IL-9 expression and disease progression, with the better prognosis shown by patients with the highest levels of IL-9 in cancer tissues [66]. Furthermore, Th9 cells play an anti-tumoral role in breast cancer. Th9 cells, which are significantly increased in the peripheral blood of breast cancer patients [67], act via the secretion of both IL-9 and IL-21, which promotes cytotoxicity of tumor-specific CTLs [68].

As opposed to solid tumors, where IL-9 acts as a protective soluble molecule in the tumor microenvironment, IL-9 mainly exerts a pro-tumoral effect in hematologic malignancies. The pro-proliferative and anti-apoptotic activities of IL-9 in hematologic neoplasias result from its ability to trigger the JAK/STAT pathways [69], which eventually stimulate neoplastic cell accumulation and promote disease progression [17]. Moreover, IL-9 protects neoplastic cells from dexamethasone-induced apoptosis [70].

Enhanced expression of IL-9 and IL-9Rα is detectable in biopsies and sera from patients with Hodgkin’s lymphoma, anaplastic large cell lymphoma [71,72], and nasal NK/T-cell lymphoma [73,74]. Wang and colleagues reported elevated serum levels of IL-9 in B cells from non-Hodgkin’s lymphoma and diffuse large B-cell lymphoma (DLBCL) patients [75], along with high levels of IL-9R expression in tumoral tissues that correlate with adverse prognostic markers of the disease [76]. Moreover, they showed that neutralizing anti-IL-9 or anti-IL-9R antibodies significantly inhibit tumor growth in mouse models of lymphoma [75]. They more recently demonstrated that high serum levels of IL-9 in DLBCL patients correlate with prolonged survival and reduced sensitivity to chemotherapeutic drugs of neoplastic cells [77].

IL-9 also participates in the pathogenesis of T cell hematologic malignancies. As mentioned above, T-ALL cells are highly sensitive to IL-9 as a result of RasGRP1 overexpression, which promotes Ras/MAPK signaling and enhances leukemic cell proliferation and survival [45]. Moreover, the high levels of Th9-secreted IL-9 found in Cutaneous T-Cell Lymphoma also participate in tumor development by reducing the oxidative stress of leukemic cells. thereby promoting their survival [78]. Adult T-cell leukemia is an additional example of the pro-tumoral function of IL-9. In this neoplasia, the activation of IL-9-dependent autocrine/paracrine loops results in amplified JAK/STAT signaling and enhanced tumor cell survival and proliferation [79].

## 5. IL-9 Acts as a Pro-Tumoral Soluble Factor in Chronic Lymphocytic Leukemia (CLL)

Chronic lymphocytic leukemia (CLL) is a strikingly heterogeneous hematologic malignancy, both molecularly and clinically. It is characterized by the accumulation of mature monoclonal B lymphocytes with a CD19^+^/CD5^+^/CD23^+^ phenotype in the bone marrow, peripheral blood, and lymphoid tissues [80], where malignant B cells receive key proliferation and survival signals by immune and non-immune microenvironmental cells [81]. Notably, CLL B cells and the tumor microenvironment dynamically co-evolve not only through direct cell–cell contact, but also through massive and unbalanced secretion of cytokines, growth factors, and extracellular vesicles, thereby paving the way to the establishment of a pro-inflammatory and immunosuppressive microenvironment [82].

IL-2, IL-4, IL-8, IL-22, IL-23, and TNF-α all exert a frank pro-tumoral activity in CLL by stimulating STAT and NF-κB transcription factors and enhancing tumor cell proliferation [17]. By contrast, other cytokines exert dual and opposing activities. IL-6 activates the pro-survival transcription factors STAT3 and NF-κB in CLL [83], while, in contrast, suppressing toll-like receptor signaling [84]. IL-15 also harbors two different and opposing functions in CLL pathogenesis. This cytokine promotes CLL cell proliferation and prevents apoptosis induced by surface IgM cross-linking [85], but also promotes autologous NK cell proliferation and enhances the sensitivity of leukemic cells to the anti-CD20 antibody Rituximab [86,87]. The implication of IL-17 in CLL remains unclear. Its downregulation is associated with Treg expansion and disease progression in CLL [88], but at the same time, it is upregulated, along with IL-6, in sera from CLL patients [89].

We and others recently reported the overexpression of IL-9 in leukemic cells from CLL patients [14,15,90,91,92] and Eμ-TCL1 mice [15], a well-established CLL mouse model [93]. Interestingly, this enhanced expression correlates with hallmarks of aggressive disease, such as unmutated Immunoglobulin Heavy Variable (IgHV) genes, ectopic expression of ζ-associated protein of 70 kDa (ZAP-70), and high levels of the surface glycoprotein CD38 [14,15,90]. Moreover, it correlates with lower overall survival of CLL patients [15]. IL-9 is also overexpressed in sera and peripheral blood from CLL patients [92], mainly as a result of uncontrolled secretion by Th9 cells [91].

The mechanisms regulating IL-9 overexpression in leukemic cells from CLL patients are still a matter of debate. By treating MEC1, a cell line derived from a CLL patient [94], with a STAT6 specific inhibitor, Chen and colleagues demonstrated that STAT6 is implicated in the regulation of IL-9 expression [95]. In leukemic cells isolated from peripheral blood of a small cohort of patients, Chen et al. also observed abnormal STAT3 phosphorylation and found that a positive feedback loop activated by extracellular IL-9 leads to STAT3 phosphorylation, which then further enhances IL-9 expression [14]. Interestingly, STAT3 activation leads to the upregulation of miR-155 and miR-21 expression in CLL cells [96], which, in turn, promote IL-9 expression [14]. Moreover, IL-9 expression is regulated by NF-κB in T cells [97] and mast cells [98], suggesting that a similar mechanism is operational in B cells. It is noteworthy that NF-κB is constitutively activated in CLL B cells [99], further substantiating this hypothesis (Figure 2).

A recent report by Sabry and colleagues showed that high levels of circulating IL-9 and Th9 cells strongly correlate with oxidative stress in leukemic cells from CLL patients, which, in turn, correlates with markers of unfavorable prognosis, such as abnormal immunophenotype and cytogenetic aberrations. These results strongly suggest the existence of a link between Th9-secreted IL-9 and oxidant-dependent injury in CLL B cells, although the underlying molecular mechanism remains to be clarified [91].

We recently discovered that an additional mechanism is implicated in IL-9 overexpression in leukemic cells from CLL patients. We found that IL-9 overexpression strongly correlates with the expression defect of the proapoptotic adaptor p66Shc that has been associated with CLL [15,100]. The strong inverse relationship between p66Shc and IL-9 expression was demonstrated by the drop in IL-9 expression in CLL B cells where p66Shc expression was reconstituted by transient transfection [15]. IL-9 overexpression was also observable in leukemic cells from Eμ-TCL1 with overt leukemia, along with a profound decrease in p66Shc expression, and further increased in leukemic cells from Eμ-TCL1/p66Shc^−/−^ mice [15], demonstrating that p66Shc participates in the control of IL-9 expression (Figure 2). The molecular mechanism linking p66Shc to IL-9 expression is, as yet, unknown, although we can hypothesize a role for the well-known pro-oxidant activity of p66Shc. The p66Shc defect observed in CLL was indeed found to be related to a substantial decrease in intracellular reactive oxygen species (ROS) [101]. The fact that IL-9 expression decreases in CLL B cells reconstituted with wild-type p66Shc but not with a p66Shc mutant lacking the ROS-elevating activity [15] strongly supports the hypothesis that a still-unknown ROS-sensitive transcription factor might be responsible for the enhanced expression of IL-9.

These findings indicate that, in the specific context of CLL, IL-9 is overexpressed not only by Th9 cells, but also by leukemic cells themselves. Irrespective of the cells involved in the secretion of this cytokine, the increased amount of IL-9 in the CLL tumor microenvironment has been proven to promote tumor development [14,15,90,91]. To date, the effects of this cytokine on the immune components of the tumor microenvironment have not been investigated. Interestingly, we showed that IL-9 acts on the stromal cells of lymphoid organs by enhancing their secretion of the homing chemokines CCL2, CXCL13, CCL21, and CXCL9, -10, and -11, ligands of the chemokine receptors CCR2, CXCR5, CCR7, and CXCR3, respectively [15]. We correlated the enhanced release of homing chemokines by stromal cells to the enhanced accumulation of CLL cells in the pro-survival and chemoprotective lymphoid niche [80] (Figure 2). Leukemic cells from Eμ-TCL1/p66Shc^−/−^ mice, that secrete even higher amounts of IL-9 compared to their Eμ-TCL1 counterparts, also harbor a higher rate of homing to lymphoid organs [101], which correlates to the enhanced secretion of homing chemokines by stromal cells [15]. Of note, while both CXCL13 and CCL21 typically control lymphocyte homing to secondary lymphoid organs, the CCL2 and the CXCR3 ligands CXCL9, -10, and -11 have been found to guide lymphocyte homing toward extranodal sites such as the liver and lung, which are colonized by leukemic cells in CLL patients with highly aggressive disease presentation [102,103,104]. Indeed, Eμ-TCL1/p66Shc^−/−^ mice show a profound structural alteration of both the liver and lung as a result of high leukemic cell homing to these sites [101]. Hence, while the potential effects of IL-9 dysregulation on other components of the tumor microenvironment remain to be clarified, we can definitely qualify IL-9 as a pro-tumoral cytokine in the CLL context, with both autocrine pro-survival effects on leukemic cells, indirect pro-chemotactic effects through paracrine conditioning of neighboring stromal cells, and long-distance effects on stromal cells far from the lymphoid tumor microenvironment in order to generate new leukemic foci.

## 6. Conclusions

Accumulating evidence establishes IL-9 as a key player in disease pathogenesis, albeit with different and sometimes opposing activities in specific disease contexts. Here, we reviewed our current understanding of the role of IL-9 in several neoplasias, and pointed out that it exerts protective activities on solid tumors while acting as a pro-tumoral soluble factor in hematologic malignancies. It is noteworthy that solid and hematologic tumors show remarkable differences in their tumor architecture. Indeed, in solid tumors, neoplastic cells locate in the center, and the surrounding tumor microenvironment forms a barrier that hampers immune cell infiltration, thereby protecting tumor cells from elimination. By contrast, in hematologic malignancies tumor cells intimately associate with cellular infiltrates of the tumor microenvironment [8]. Hence, the tight dialogue between neoplastic cells and the microenvironment in hematologic malignancies turns into the minimal and less efficient dialogue in solid malignancies. This crucial “anatomical” difference might account for the opposing functions of IL-9 in neoplasias, and suggests potentially different outcomes of anti-cancer therapies based on anti-IL-9 antibodies. It is noteworthy that additional careful evaluation of IL-9 secretion in both solid and hematologic malignancies is required before considering its potential use as an anti-cancer therapeutic.

CLL is a hematologic malignancy where IL-9 has pro-tumoral functions. Secreted by both Th9 cells [91] and leukemic cells themselves, it activates JAK/STAT-dependent signals, which promote neoplastic cell survival and proliferation [14,92]. Moreover, it stimulates stromal cells to secrete homing chemokines, ultimately enhancing the ability of leukemic cells to home to the lymphoid stroma, thereby further promoting their survival [15]. Targeted therapies that are able to reduce IL-9 in CLL patients might, therefore, counteract its detrimental effects in CLL pathogenesis. Intravenous injection of monoclonal anti-IL-9 antibodies was proven effective in ameliorating disease outcomes in the Eμ-TCL1/p66Shc^−/−^ mouse model of aggressive CLL [15], paving the way to further pre-clinical studies. In the era of targeted therapies, the strong association between circulating IL-9 and markers of unfavorable CLL prognosis [14,15,90] suggests the potential use of anti-IL-9 antibodies as a therapeutic option for patients with high IL-9 levels. Interestingly glucocorticoids, potent immune-suppressive agents that reduce cytokine expression by inhibiting transcription factors such as Activation Protein (AP)-1 and NF-κB, have been observed to reduce IL-9 expression in asthma patients [40,105]. However, they also reduce the expression of other cytokines, including IL-5 and IL-13, as a consequence of a shared regulatory mechanism [105], making this therapeutic hypothesis hardly feasible.

CAR-T cell therapy is effective for hematologic malignancies. However, no approved CAR-T cell therapies for CLL are available yet [106]. Interestingly, a recent existing finding demonstrates that human CAR-T cells polarized and expanded under a Th9-culture condition (T9 CAR-T) display stronger anti-tumor activity against established CD19-expressing human ALL or GPC3-expressing liver carcinoma in vivo compared to the classical IL-2-polarized (T1) CAR-T cells [107]. Compared to the T1 subtype, T9 CAR-T cells preferentially secrete IL-9, which displays anti-tumoral effects on both hematologic (ALL) and solid (liver carcinoma) tumors. This result further highlights that our knowledge of IL-9 behavior in tumor control is still far from complete but also underscores the importance of investigating the full array of tumor-specific activities of this cytokine.

The role of Tc9 cells in CLL has not yet been addressed. Interestingly, while, in melanoma models, Tc9 cells have a strong and persistent anti-tumor effect that mainly depends on the production of IL-9 [108], Tc9 cells show a weak cytolytic ability in vitro [28,109,110]. Given that T cell-mediated cytotoxicity is suppressed in CLL [111], studies focused on this immune cell population might provide new clues to restore CTL-mediated anti-tumor responses in CLL.

## Figures and Tables

**Figure 1 cancers-13-06301-f001:**
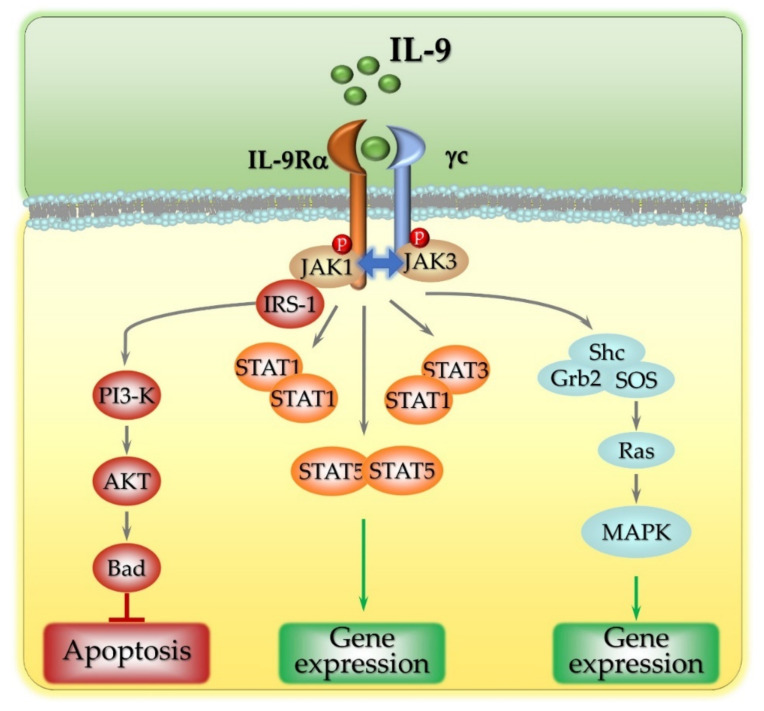
IL-9 Signaling Pathways. IL-9 binds to the heterodimeric receptor IL-9R and recruits the Janus kinases JAK1 and JAK3. The phosphorylated JAKs activate the STAT1 and STAT5 homodimers and the STAT1:STAT3 heterodimer, which translocate into the nucleus and bind DNA to control gene expression. JAK1 also activates IRS-1, which, in turn, activates the PI3-K/Akt-dependent anti-apoptotic pathway, and the Ras/MAPK signaling pathway that controls gene expression.

**Figure 2 cancers-13-06301-f002:**
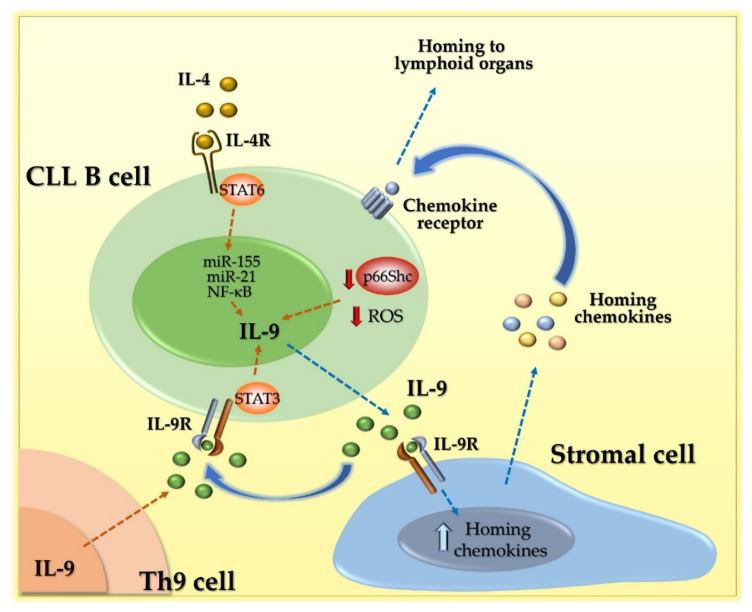
IL-9 expression and functions in CLL. IL-9 expression in CLL B cells is positively regulated in the following ways: (i) by the binding of IL-4 to the IL-4 receptor IL-4R and STAT6 activation; (ii) by the defective expression of p66Shc and the resulting low levels of intracellular ROS; and (iii) by a positive feedback loop activated by IL-9 secreted by CLL B cells themselves, which binds IL-9R and activates STAT3. IL-9 also acts on stromal cells, enhancing the expression of homing chemokines that, in turn, act on CLL B cells to promote their homing to lymphoid organs. ROS: reactive oxygen species.

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
