# Peer review of "Interleukin (IL)-9 Supports the Tumor-Promoting Environment of Chronic Lymphocytic Leukemia"

_cancers, 2021, doi:10.3390/cancers13246301_

Round 1
Reviewer 1 Report
Good review of current state of knowledge.
EXcellent graphics for improved clarity.
Would benefit from some speculation as to where this field might develop, and how this knowledge might play a role in therapeutic advances,
Author Response
According to the Reviewer’s suggestion, we included a comment on the use of IL-9 as potential anti-cancer approach in the Conclusions paragraph (please see lines 365-368 and 378-380 of the revised manuscript).
Reviewer 2 Report
This review presents a comprehensive summary of the effects of IL-9 on tumor progression and anti-tumor immunity. It is well written and organized, adequately referenced and provides extensive information on a new and interesting disease-relevant factor. It can be published in its current form.
Some minor typos or language issues:
line 48: which are breast, ...
line 70: in the late 80s
line 102: signaling molecule is shared
line 243: "contributes to" or "participates in"
line 247: see line 243
lines 266,268,269: harbors, promotes, enhances
line 271: but at the same time
line 274: better: an well established CLL mouse model
Author Response
We thank the Reviewer for his/her suggestions concerning language. We changed the main text accordingly.
Reviewer 3 Report
Patrussi and coworker set out to summarize the role of IL-9 in the tumor-promoting environment of CLL
The review article is well organized and clearly written.
Because high levels of circulating IL-9 is associated with markers of unfavourable prognosis, the authors should briefly discuss the potential role of IL-9 in the era of target therapy (BTK and BCL-2 inhibitors).
Author Response
According to the Reviewer’s suggestion, we added a sentence in the Conclusions paragraph that comments on the potential use on anti-IL-9 antibodies as targeted therapy for the treatment of CLL patients with elevated IL-9 levels (please see lines 378-380).